# Using Multimodal Foundation Models and Clustering for Improved Style Ambiguity Loss

## Abstract

Teaching text-to-image models to be creative involves using style ambiguity loss, which requires a pretrained classifier. In this work, we explore a new form of the style ambiguity training objective, used to approximate creativity, that does not require training a classifier or even a labeled dataset. We then train a diffusion model to maximize style ambiguity to imbue the diffusion model with creativity and find our new methods improve upon the traditional method, based on automated metrics for human judgment, while still maintaining creativity and novelty.

## 1 Introduction

With every new invention comes a new wave of possibilities. Humans have been making pictures since before recorded history, so its only natural that there would be interest in computational image generation. Artificially generating photographs that are indistinguishable from real ones has become so easy and effective that there is even concern over "deepfakes" being used for propaganda or illicit purposes (Pawelec, 2022). On the other hand, generating images that look like art is a slightly different problem. The exact mathematical properties of what constitutes "quality" art is not as easy to quantify as other tasks, like classification accuracy, prediction error or whether a question was answered correctly. While machines can very easily be trained to mimic a dataset, humans like to be surprised by novelty, without feeling like they are being exposed to total randomness. A breakthrough was the invention of the Creative Adversarial Network (Elgammal et al., 2017), which used a style ambiguity loss to train a network to generate images that could not be classified as belonging to a particular style. However, GANs have largely been superseded by diffusion models (Luo, 2022), due to their far better results. Additionally, the style ambiguity loss requires a pretrained classifier. Every set of styles or concepts requires training a classifier before even training a model to generate images. Furthermore, training a classifier requires that the dataset be labeled correctly, and manually labeling a dataset is often even more expensive and time-consuming than training a model. To circumvent these issues, we propose using a classifier that does not require any additional training and can be easily applied to any dataset, labeled or unlabeled. Our contributions are as follows:

- We applied creative style ambiguity loss to diffusion models, which are easier to train and produce higher-quality images than GANs.

- We developed versatile CLIP-based and K-Means-based creative style ambiguity losses that do not require training a separate GAN-based style classifier.

- Empirically, we find our new creative style ambiguity loss can be used to tune a diffusion model to generate samples that are higher quality than the generated samples of a diffusion model trained with the pre-existing GAN-based style ambiguity loss

## 2 Related Work

### 2.1 Creativity

Creativity has been hard to define and quantify. Creative work has been formulated as work having novelty, in that it differs from other similar objects, and also utility, in that it still performs a function (Cropley, 2006). For example, a Corinthian column has elaborate, interesting, unexpected adornments (novelty) but still holds up a building (utility). A distinction can also be made between "P-creativity", where the work is novel to the creator, and "H-creativity" where the work is novel to everyone (Boden, 1990). Computational techniques to be creative include using genetic algorithms (DiPaola & Gabora, 2008), reconstructing artifacts from novel collections of attributes (Iqbal et al., 2016), and most relevantly to this work, using Generative Adversarial Networks (Elgammal et al., 2017) with a style ambiguity loss.

### 2.2 Computational Art

One of the first algorithmic approaches dates back to the 1970s with the now primitive AARON (McCorduck, 1991), which was initially only capable of drawing black and white sketches. Generative Adversarial Networks (Goodfellow et al., 2014), or GANs, were some of the first models to be able to create complex, photorealistic images and seemed to have potential to be able to make art. Despite many problems with GANs, such as mode collapse and unstable training (Saxena & Cao, 2023), GANs and further improvements (Arjovsky et al., 2017; Karras et al., 2019; 2018) were state of the art until the introduction of diffusion Sohl-Dickstein et al. (2015). Diffusion models such as IMAGEN (Saharia et al., 2022) and DALLE-3 (Betker et al.) have attained widespread commercial success (and controversy) due to their widespread adoption.

### 2.3 Reinforcement Learning

Reinforcement learning (RL) is a method of training a model by having it take actions that generate a reward signal and change the environment, thus changing the impact and availability of future actions Qiang & Zhongli (2011). RL has been used for tasks as diverse as playing board games (Silver et al., 2017), protein design (Lutz et al., 2023), self-driving vehicles (Kiran et al., 2021) and quantitative finance (Sahu et al., 2023). Policy-gradient RL (Sutton et al., 1999) optimizes a policy $\pi$ that chooses which action to take at any given timestep, as opposed to value-based methods that may use a heuristic to determine the optimal choice. Examples of policy gradient methods include Soft Actor Critic (Haarnoja et al., 2018), Deep Deterministic Policy Gradient (Lillicrap et al., 2019) and Trust Region Policy Optimization (Schulman et al., 2017a).

## 3 Method

### 3.1 Model

#### 3.1.1 Creative Adversarial Network

A Generative Adversarial Network, or GAN (Goodfellow et al., 2014), consists of two models, a generator and a discriminator. The generator generates samples from noise, and the discriminator detects if the samples are drawn from the real data or generated. During training, the generator is trained to trick the discriminator into classifying generated images as real, and the discriminator is trained to classify images correctly. Given a generator $G : \mathbb{R}^{noise} \to \mathbb{R}^{h \times w \times 3}$, a discriminator $D : \mathbb{R}^{h \times w \times 3} \to [0, 1]$ real images $x \in \mathbb{R}^{h \times w \times 3}$, and noise $\mathcal{Z} \in \mathbb{R}^{noise}$, the objective is:

$$\min_G \max_D \mathbb{E}_x[log(D(x)] + \mathbb{E}_{\mathcal{Z}}[log(1 - D(G(\mathcal{Z})))]$$

At inference time, the generator is used to generate realistic samples. Elgammal et al. (2017) introduced the Creative Adversarial Network, or CAN, which was a DCGAN (Radford et al., 2016) where the discriminator was also trained to classify real samples, minimizing the style classification loss. Given $N$ classes of image (such as ukiyo-e, baroque, impressionism, etc.), the classification modules of the Discriminator

$D_C : \mathbb{R}^{h \times w \times 3} \to \mathbb{R}^N$ that returns a probability distribution over the $N_s$ style classes for an image and the real labels $\ell \in \mathbb{R}^N$, the style classification loss was:

$$L_{SL} = \mathbb{E}_{x,\ell}[\mathbf{CE}(D_C(x), \ell)]$$

Where $\mathbf{CE}$ is the cross entropy function.

The generator was also trained to generate samples that could not be easily classified as belonging to one class. This stylistic ambiguity is a proxy for creativity or novelty. Given a vector $U \in \mathbb{R}^N$, where each entry $u_1, u_2,,, u_N = \frac{1}{N}$, and some classifier $C : \mathbb{R}^{h \times w \times 3} \to \mathbb{R}^N$ the style ambiguity loss is:

$$L_{SA} = \mathbb{E}_{\mathcal{Z}}[\mathbf{CE}(C(G(\mathcal{Z})), U)]$$

The discriminator was additionally trained to minimize $L_{SL}$ and the generator was additionally trained to minimize $L_{SA}$. In the original work, the authors set $C = D_C$. For our work, we will be combining the Wasserstein and CAN methods. We used the following loss functions:

$$L_{disc} = \mathbb{E}_x[log(D(x)] + \mathbb{E}_{\mathcal{Z}}[log(1 - D(G(\mathcal{Z}))] + L_{SL}$$

$$L_{gen} = -\mathbb{E}_x[log(D(x)] - \mathbb{E}_{\mathcal{Z}}[log(1 - D(G(\mathcal{Z}))] + L_{SA}$$

### 3.1.2 Diffusion

A diffusion model aims to learn to iteratively remove the noise from a corrupted sample to restore the original. Starting with $x_0$, the forward process $q$ iteratively adds Gaussian noise to produce the noised version $x_T$, using a noise schedule $\beta_1...\beta_T$, which can be learned or manually set as a hyperparameter:

$$q(x_{1:T}|x_0) = \prod_{t=1}^{T} q(x_t|x_{t-1})$$

$$q(x_t|x_{t-1}) = \mathcal{N}(x_t; \sqrt{1 - \beta_t}x_{t-1}, \beta_t\mathbf{I})$$

More importantly, we also want to model the reverse process $p$, that turns a noisy sample $x_T$ back into $x_0$, conditioned on some context $c$. As $x_T$ is the fully noised version, $p(x_T|c) = \mathcal{N}(x_T; \mathbf{0}, \mathbf{I})$

$$p_\theta(x_{0:T}|c) = p(x_T|c) \prod_{t=1}^{T} p_\theta(x_{t-1}|x_t, c)$$

$$p_\theta(x_{t-1}|x_t, c) = \mathcal{N}(x_{t-1}; \mu_\theta(x_t, t, c), \Sigma_\theta(x_t, t, c))$$

We train $\Sigma_\theta$ and $\mu_\theta$ via optimizing the variational lower bound of the negative likelihood of the data:

$$\mathbb{E}[-\mathrm{log}p_\theta(x_0)] \leq \mathbb{E}[-\mathrm{log}\frac{p_\theta(x_{0:T}|c)}{q(x_{1:T}|x_0)} = L$$

As shown by Ho et al. (2020), this is equivalent to estimating the noise at each step using a model $\epsilon_\theta$. So the loss to be optimized is:

$$L = \mathbb{E}_{x,\epsilon \sim \mathcal{N}(0,1),t}||\epsilon - \epsilon_\theta(x_t, t)||_2^2$$

Once the model has been trained, the reverse process, aka inference, to generate a sample from noise $x_T \sim \mathcal{N}(0,1)$ can be done iteratively by finding $x_t - 1$ given $x_t, \alpha_t = 1 - \beta_t, \bar{\alpha}_t = \prod_s^t \alpha_s, \mathcal{Z} \sim \mathcal{N}(0,1)$ and $\sigma_t^2 = \beta_t$ or $\sigma_t^2 = \frac{1-\alpha_{t-1}}{1-\alpha_t}\beta_t$ :

$$x_{t-1} = \frac{1}{\sqrt{\alpha_t}}(x_t - \frac{1 - \alpha_t}{\sqrt{1 - \bar{\alpha}_t}}\epsilon_\theta(x_t, t)) + \sigma_t\mathcal{Z}$$

Acording to Ho et al. (2020), both versions of $\sigma_t$ had similar results. In our case, we used $\sigma_t^2 = \frac{1-\alpha_{t-1}}{1-\alpha_t}\beta_t$.

A Variational Autoencoder (Kingma & Welling, 2022) consists of an encoder $\mathcal{E} : \mathbb{R}^{h \times w \times 3} \to \mathbb{R}^{h_z \times w_z \times c_z}$ to map an image into a lower-dimensional latent space, and a decoder $\mathcal{D} : \mathbb{R}^{h_z \times w_z \times c_z} \to \mathbb{R}^{h \times w \times 3}$ to reverse this process. Rombach et al. (2022) performs diffusion but uses the latent representation of images $z_0 = \mathcal{E}(x_0)$:

$$L = \mathbb{E}_{x, \epsilon \sim \mathcal{N}(0,1), t} ||\epsilon - \epsilon_\theta(z_t, t)||_2^2$$

This method, which we employed in this work, is known as stable diffusion. The encoding and decoding between the image dimensions and the latent dimensions is often implicit, and for the rest of the paper we will use $x_t$ not $z_t$, as is common in the literature.

### 3.1.3 Markov Decision Processes

A Markov Decision Process (Bellman, 1957) is defined as a tuple $(S, A, p_0, P, R)$ that models the actions of an agent in some environment with discrete time-steps.

$S$ is the state space, the set of states the environment can be in.

$A$ is the set of actions that the agent can take.

$p_0$ is the initial distributions of states $s \in S$ when $t = 0$.

$P_a(s, s')$ is the probability of transitioning from state $s$ at time $t$ to $s'$ at $t + 1$ when the agent has taken action $a \in A$.

The reward function $R(s_t, a_t)$ returns a reward a time $t$ given the action $a_t$ the agent takes and the state of the environment $s_t$.

The agents actions are determined by the policy $\pi(a|s)$ that maps actions to states. The series of state-action pairs for each timestep is called a trajectory $\tau = (s_0, a_0 ....s_T, a_T)$. Using policy-gradient as opposed to value-based RL, we train $\pi$ by maximizing the reward $R$ over the trajectories sampled from the policy:

$$\mathcal{J}_{RL}(\pi) = \mathbb{E}_{\tau \sim p(\tau|\pi)}[\sum_{t=0}^{T} R(s_t, a_t)]$$

### 3.1.4 Denoising Diffusion Proximal Optimisation

Introduced by Black et al. (2023), Denoising Diffusion Proximal Optimisation, or DDPO, represents the Diffusion Process as a Markov Decision Process. A similar method was also pursued by Fan et al. (2023).

$$a_t \triangleq x_{t-1}$$

$$s_t \triangleq (c, t, x_t)$$

$$\pi(a_t|s_t) \triangleq p_\theta(x_{t-1}|x_t, c)$$

$$p_0(s_0) \triangleq (p(c), \delta_T, \mathcal{N}(0, \mathbf{I}))$$

$$P(s_{t+1}) \triangleq (\delta_c, \delta_{t-1}, \delta_{x_{t-1}})$$

$$R(s_t, a_t) \triangleq r(x_0, c)$$

$$\mathcal{J}_{RL}(\pi) \triangleq \mathcal{J}_{DDRL}(\theta) = \mathbb{E}_{c \sim p(c), x_0 \sim p_\theta(x_0|c)}[r(x_0, c)]$$

Reinforcement learning training was then applied to a pretrained diffusion model, which in our case was Stable Diffusion 2 (Rombach et al., 2022). Following Schulman et al. (2017b), Black et al. (2023) also implemented clipping to protect the policy gradient $\nabla_\theta \mathcal{J}_{DDRL}$ from excessively large updates. We largely follow their method but use a different reward function. We fine-tune off of the pre-existing **stabilityai/stable-diffusion-2-base** checkpoint (Rombach et al., 2022) downloaded from `https://huggingface.co/stabilityai/stable-diffusion-2-base`.

## 3.2 Reward Function

In the original paper, the authors used four different reward functions for four different tasks. For example, they used a scorer trained on the LAION dataset (Schuhmann & Beaumont, 2022) as the reward function to improve the aesthetic quality of generated outputs. In this paper, we use the reward model based on Elgammal et al. (2017), where the model is rewarded for stylistic ambiguity. Given a generated image $x_0 \in \mathbb{R}^{h \times w \times 3}$ and a classifier $C : \mathbb{R}^{h \times w \times 3} \to \mathbb{R}^N$ we want to maximize:

$$R(x_0) = -\mathbf{CE}(C(x_0), U)$$

where **CE** is the cross entropy.

## 3.3 Data

Starting with the WikiArt dataset (Saleh & Elgammal, 2015), we used 1000 images from each class, oversampling when necessary, to balance the distributions between classes, to train the CAN. To train the diffusion model, we prompted the model by concatenating a randomly selected medium prompt from (**painting of** , **picture of** , **drawing of** ) to a randomly selected subject prompt (**a man**, **a woman**, **a landscape**, **nature**, **a building**, **an animal**, **shapes**,  **an object**). An example prompt would be **picture of an animal**. With 10% probability we would set the prompt to the null string in order to train the model unconditionally as well.

## 3.4 Choice of Classifier

Style ambiguity loss relies on some classifier $C$. We are exploring four versions of this classifier.

### 3.4.1 DCGAN-Based Classifier

We can use the classification module of the discriminator as the classifier in the reward function, setting $C = D_C$. In the case of the CAN, $D_C$ is trained jointly along with the generator. In the case of DDPO, we use a pretrained $D_C$ from the CAN discriminator (which we call Diffusion DCGAN Based), or a pretrained $D_C$ that was trained simply to classify samples as belonging to a particular class (which we call Diffusion Simple Classifier Based), without the use of a generator or any GAN training objectives.

### 3.4.2 CLIP-Based Classifier

Given text $\in \mathbb{R}^{text}$ and an image $\in \mathbb{R}^{h \times w \times 3}$,we can use a pretrained CLIP (Radford et al., 2021) model, that can return a similarity score for each image-text pair: $CLIP : \mathbb{R}^{text} \times \mathbb{R}^{h \times w \times 3} \to \mathbb{R}$. CLIP is a multimodal foundation model trained using contrastive learning (Jaiswal et al., 2021) on a dataset of approximately 400 million text-image pairs. For each generated image $x_0$, for each class name $s_i, 1 \le i \le N_s$, we find $CLIP(s_i, x_0)$. We can then create a vector $(CLIP(s_1, x_0), CLIP(s_2, x_0), , , CLIP(s_{N_s}, x_0))$ and then use softmax to normalize the vector and define the result as $C_{CLIP}(x_0)$. Formally:

$$C_{CLIP}(x_0) = \mathbf{softmax}((CLIP(s_1, x_0), CLIP(s_2, x_0), , , CLIP(s_{N_s}, x_0))$$

Then we set $C = C_{CLIP}$. We discard the results of $D_C$ when using a CLIP-Based Classifier with CAN. We used the 27 style classes in the WikiArt dataset (Saleh & Elgammal, 2015) as $s_i, 1 \le i \le N_s$. A list of said classes can be found in Appendix B. We used the **clip-vit-large-patch14** CLIP checkpoint downloaded from `https://huggingface.co/openai/clip-vit-large-patch14`.

### 3.4.3 K-Means Text and Image Based Classifiers

Alternatively, when we have $N_s$ text labels or $N_I$ source images, we can embed the labels or images into the CLIP embedding space $\in \mathbb{R}^{768}$ and perform k-means clustering to generate k centers. Given a CLIP Embedder $E : \mathbb{R}^{h \times w \times 3} \to \mathbb{R}^{768}$ mapping images to embeddings, and the k centers $c_1, c_2, , , c_k$ we can create

a vector $(\frac{1}{||E(x_0)-c_1||}, \frac{1}{||E(x_0)-c_2||}, , , , \frac{1}{||E(x_0)-c_k||}$ and then use softmax to normalize the vector and define the result as $C_{KMEANS}$. Formally:

$$C_{KMEANS}(x_0) = \mathbf{softmax}(\frac{1}{||E(x_0)-c_1||}, \frac{1}{||E(x_0)-c_2||}, , , , \frac{1}{||E(x_0)-c_k||})$$

Then we set $C = C_{KMEANS}$. We used two sets of centers: one set from performing k-means clustering on the WikiArt images, which we called **K Means Image Based**, and one set from clustering the names of the 27 style classes, which we call **K Means Text Based**.

## 4 Results

We generated all images with width and height = 512. The authors used width and height = 256 in the original CAN paper. However, given that larger, more detailed images are preferred by most people, we thought it more relevant to focus on larger images. For results on smaller images, refer to appendix C. Table 1 shows a few DDPO images with the prompts used to generate them. Appendix A shows more examples generated using different prompts.

### 4.1 Quantitative Evaluation

We generated 100 images using the same prompts the models were trained on for each model. We used three quantitative metrics to score the models

- **AVA Score:** Consisting of CLIP+Multi-Layer Perceptron (Haykin, 2000), the AVA model was trained on the AVA dataset (Murray et al., 2016) of images and average rankings by human subjects, in order to learn to approximate human preferences given an image. We used the CLIP model weights from the **clip-vit-large-patch14** checkpoint and the Multi-Layer Perceptron weights downloaded from `https://huggingface.co/trl-lib/ddpo-aesthetic-predictor`.

- **Image Reward:** The image reward model (Xu et al., 2023) was trained to score images given their text description based on a dataset of images and human rankings. We used the **image-reward** python library found at `https://github.com/THUDM/ImageReward/tree/main`.

- **Prompt Similarity:** Given the CLIP model's ability to embed images and text into the same space, we can measure the similarity between an image and its source prompt by finding the cosine similarity between the two CLIP embeddings. We used the **clip-vit-large-patch14** checkpoint.

Results of our experiments are shown in table 2. The best scores are bolded. There was little variance in prompt similarity. However, both K-Means-based approaches improved upon the DCGAN-based approach in terms of the two metrics for human preferences, showing that our method improves upon the past work aesthetically while also circumventing the costly training time of using a CAN or needing a labeled dataset for training the style classifier component of the CAN.

### 4.2 Comparison with Baseline

It is worth contrasting our trained DDPO model with the default pretrained **stabilityai/stable-diffusion-2-base** checkpoint diffusion model we are fine-tuning (Rombach et al., 2022). This allows us to better visualize what difference the DDPO training with style ambiguity loss actually makes. We assumed that the DDPO images might be similar to the baseline model images generated with fewer steps, so we compared DDPO Images to the baseline using 30,15 and 10 inference steps, as seen in figure 3.

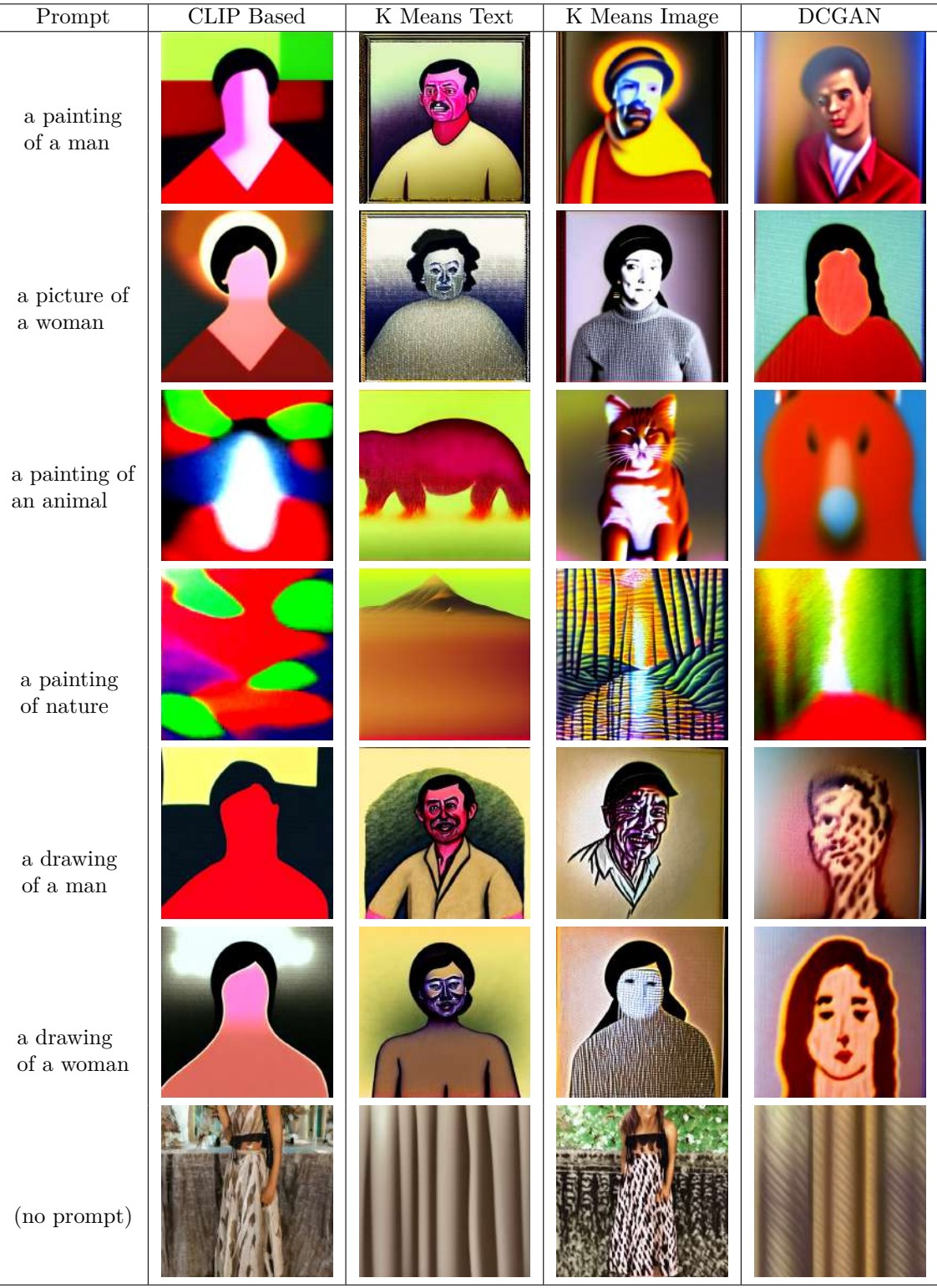

Table 1: Example Images

In order to quantitatively compare our models to the baselines, we used the embedding of the [**CLS**] token from a vision transformer loaded from the **dino-vits16** checkpoint (Caron et al., 2021) from https://

| Model | AVA Score | Image Reward | Prompt Similarity |
|---|---|---|---|
| Diffusion- CLIP Based | 4.18 | -1.75 | 0.24 |
| Diffusion- K-Means Text Based | **4.60** | -1.21 | **0.26** |
| Diffusion- K-Means Image Based | 4.38 | **-0.90** | **0.26** |
| Diffusion- DCGAN Based | 4.26 | -1.58 | **0.26** |

Table 2: Scores (Image Dim 512)

| Prompt | CLIP Based | K Means Text | K Means Image | DCGAN | Baseline (30 Steps) | Baseline (15 Steps) | Baseline (10 Steps) |
|---|---|---|---|---|---|---|---|
| a painting of a landscape | | | | | | | |
| a picture of an animal | | | | | | | |
| a painting of a man | | | | | | | |

Table 3: Example Images

`huggingface.co/facebook/dino-vits16`, as that encodes stylistic information (Tumanyan et al., 2022; Kwon & Ye, 2023). We averaged the cosine similarity between style embeddings of each pair of images $(x, y_{30}, y_{15}, y_{10})$, where $x$ was generated by the tuned model and $y_{30}, y_{15}, y_{10}$ was generated by the baseline model using 30,15 and 10 inference steps respectively, using the same prompt and initial random seed. We did this 40 times. Average style cosine similarities between the DDPO-trained models and the baselines are shown in table 4. A lower style cosine similarity implies that the diffusion model has learned to successfully "deviate" from the baseline. All diffusion models model were more similar to the 30-step baseline, which implies that the diffusion models are not just learning to generate blurrier, less precise samples.

| | Baseline 30 Steps | Baseline 15 Steps | Baseline 10 Steps |
|---|---|---|---|
| Diffusion- CLIP Based | 0.29 | 0.27 | 0.23 |
| Diffusion- K-Means Text Based | 0.29 | 0.28 | 0.25 |
| Diffusion- K-Means Image Based | 0.30 | 0.30 | 0.29 |
| Diffusion- DCGAN Based | 0.26 | 0.23 | 0.21 |

Table 4: Style Similarities

# 5 Conclusion

Training models with stylistic ambiguity loss teaches them to be creative. This work introduces new forms of stylistic ambiguity loss that do not require training a classifier or GAN, which can be time-consuming and unstable (Saxena & Cao, 2023). These new methods, particularly the K-Means-based approaches, scored higher than the traditional method on quantitative metrics of human judgement. Nonetheless, there are still more directions for this to go. Both the CLIP-based and K-Means Text-based style ambiguity losses require users to heuristically choose a set of styles to "deviate" from. In this work, we only used the 27 categories in the WikiArt dataset to be comparable to the original CAN paper. However, users may instead prefer a different set of styles or words, which may produce better or more interesting results. Additionally, the K-Means Image-based style ambiguity loss does not require a multimodal model like CLIP. We could have used any pretrained model to embed images into a lower-dimensional manifold, or trained a new one. Ergo, the K-means technique could be used for any medium, such as music (Elgammal, 2022; Zhang et al., 2023), new proteins (Winnifrith et al., 2023), stories (Mori et al., 2022) and videos (Cho et al., 2024).

**Broader Impact Statement**

Many are concerned about the impacts of generative AI. By making art, this work infringes upon a domain once exclusive to humans. Companies have faced scrutiny for possibly using AI (Gutierrez, 2024), and many creatives, such as screenwriters and actors, have voiced concerns about whether their jobs are safe (del Barco, 2023). Nonetheless, using AI can help humans by making them more efficient, providing inspiration, and generating ideas (Fortino, 2023; Campitiello, 2023; Darling, 2022). It's also not certain how copyright protection will function for AI-generated art (Watiktinnakorn et al., 2023), given copyright law is based on the premise that creative works originate solely from human authorship. Clear, consistent policies, both at the government level and by industry and/or academic groups, will be needed to mitigate the harm and maximize the benefits for all members of society.

# 6 Assistance

**Author Contributions**

This section omitted while under review.

**Acknowledgements**

This section omitted while under review.

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

## A   Images

Additional text prompts and the corresponding generated images using the DDPO models can be seen in Figures 5 and 6.

## B   WikiArt Style Classes

The 27 WikiArt style classes are listed in table 7

## C   Lower Dimensional Images

All experiments and images portrayed were done using images of dimension 512. However, we also repeated the experiments using smaller images. We briefly illustrate some example images in tables 8, 9 and 10 as well as quantitative evaluations in tables C, C and C.

## D   Training

In the interest of reproducibility and transparency, the hyperparameters are listed in table 14 and table 15. All experiments were implemented in Python, building the models in **pytorch** (Paszke et al., 2017) using **accelerate** (Gugger et al., 2022) for efficient training. The diffusion models also relied on the **trl** (von Werra et al., 2020), **diffusers** (von Platen et al., 2022) and **peft** (Mangrulkar et al., 2022) libraries. The K-Means clustering was done using the k means implementation from **scikit-learn** (Pedregosa et al., 2011).

A repository containing all code can be found on github at `REDACTED_WHILE_UNDER_REVIEW_TO_MAINTAIN_ANONYMITY`. Each experiment was run using two NVIDIA A100 GPUs with 40 GB RAM. Training times and estimated carbon emissions (Lacoste et al., 2019) calculated with `https://mlco2.github.io/impact#compute` are shown in table 16.

## D.1 Batch Size

For all DDPO models, we used an effective batch size of 8. When generating images with height and width 64, 128 and 512, we just used a batch size of 8 without gradient accumulation (Kozodoi, 2021). Curiously, for images of height and width 256, using a batch size of 8 caused an error: **RuntimeError: CUDA error: CUBLAS_STATUS_ALLOC_FAILED when calling cublasCreate(handle)**. Thus, we opted to use a batch size of 4 with 2 gradient accumulation steps, equivalent to an effective batch size of 8, which worked. In order to investigate this error, we tried training a DDPO model with a batch size of 8 on a slower CPU, which was allocated 64 GB of memory. On the CPU, the error disappeared. We conclude that the reason for this error is dependent on how exactly variables are allocated across GPUs, but a more thorough investigation is beyond the scope of this paper.

## D.2 Architecture

For diffusion model training, the text encoder, autoencoder and unet were all loaded from `https://huggingface.co/stabilityai/stable-diffusion-2-base`. These model components were all frozen, but we added trainable LoRA weights to the cross-attention layers of the Unet. Parameter counts are shown in table 17. The diffusion model components used the same amount of parameters regardless of image size, but the generator and discriminator had more parameters as image size increased.

We used the convolutional neural network (Dumoulin & Visin, 2018) architecture described in Elgammal et al. (2017) for the CAN but had to use more/less layers to produce higher/lower dimension images. The generator takes a $1 \times 100$ gaussian noise vector $\in \mathbb{R}^{100} \sim \mathcal{N}(0, I)$ and maps it to a $4 \times 4 \times 2048$ latent space, via a convolutional transpose layer with kernel size = 4 and stride =1, followed by 3, 4, 5 or 6 transpose convolutional layers corresponding to image dimensions 64, 128, 256 and 512, each upscaling the height and width dimensions by two, and halving the channel dimension (for example one of these transpose convolutional layers would map $\mathbb{R}^{4 \times 4 \times 2048} \to \mathbb{R}^{8 \times 8 \times 1024}$) followed by batch normalization (Ioffe & Szegedy, 2015) and Leaky ReLU (Maas et al., 2013), and then one final convolutional transpose layer with output channels = 3 and tanh (Dubey et al., 2022) activation function. Diagrams of the generators with image dim 512, 256, 128 and 64 are shown in the figures 1, 2, 3 and 4 respectively.

For the discriminator, we first applied a convolution layer to downscale the input image height width dimensions by 2 and mapped the 3 input channel dimensions to 32 ($\mathbb{R}^{512 \times 512 \times 3} \to \mathbb{R}^{256 \times 256 \times 32}$) with Leaky ReLU activation. Then we had 2, 3, 4 or 5 convolutional layers corresponding to image dimensions 64, 128, 256 and 512, each downscaling the height and width dimensions by 2 and doubling the channel dimension (for example, one of these convolutional layers would map $\mathbb{R}^{256 \times 256 \times 32} \to \mathbb{R}^{128 \times 128 \times 64}$) with batch normalization and Leaky ReLU activation. Then we had two more convolutional layers, each downscaling the height and width dimensions but keeping the channel dimensions constant (using the prior layer's channel dimensions), with batch normalization and Leaky ReLU activation. The output of the convolutional layers was then flattened. The discriminator had two heads- one for style classification (determining which style a real image belongs to) and one for binary classification (determining whether an image was real or fake). The binary classification head consisted of one linear layer with one output neuron. The style classification layer consisted of 2 linear layers with LeakyReLU activation and Dropout, with output 1024 output neurons and 512 output neurons, respectively, followed by a linear layer with 27 output neurons for the 27 artistic style classes. Diagrams of discriminators with image dim 512, 256, 128 and 64 are shown in the figures 5, 6, 7, and 8, respectively.

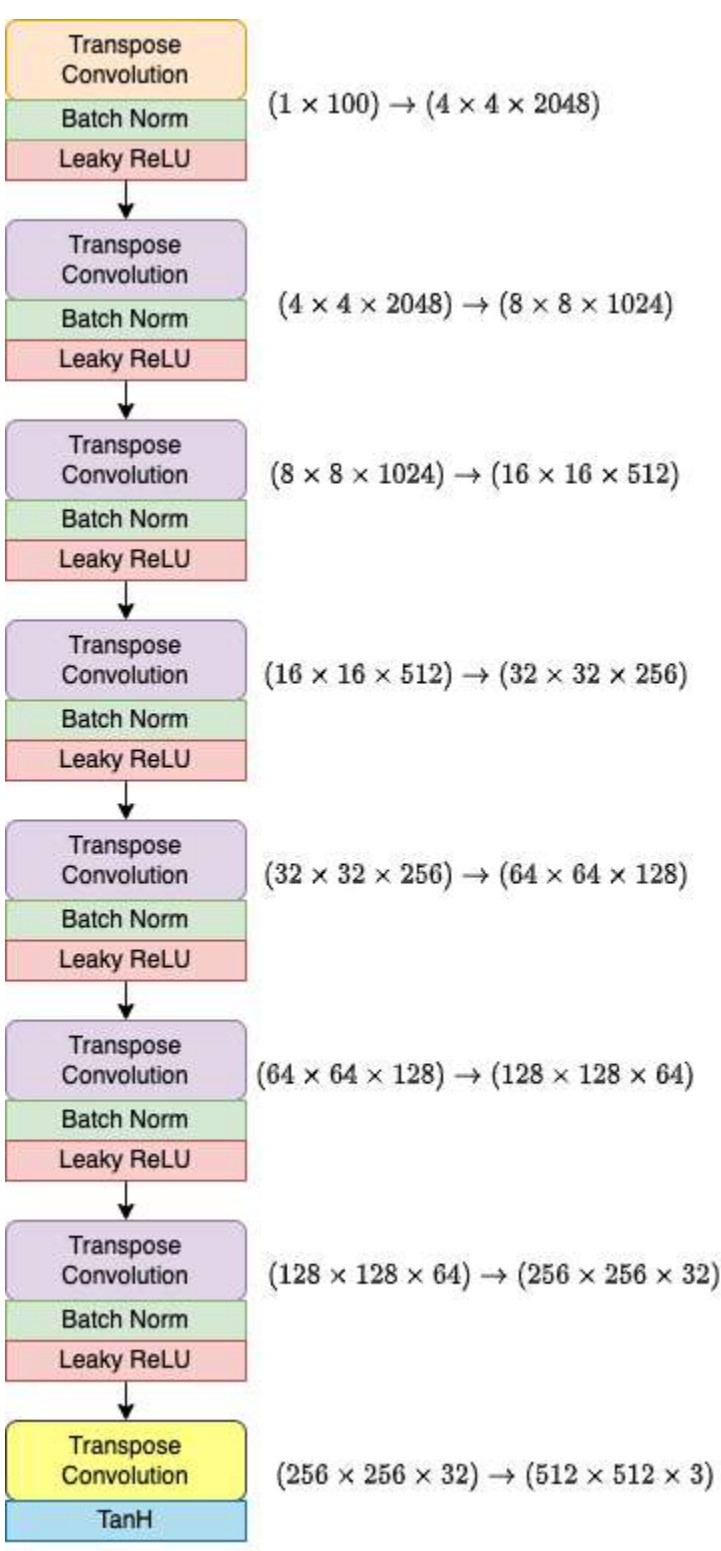

Figure 1: Generator Architecture (Image Dim 512)

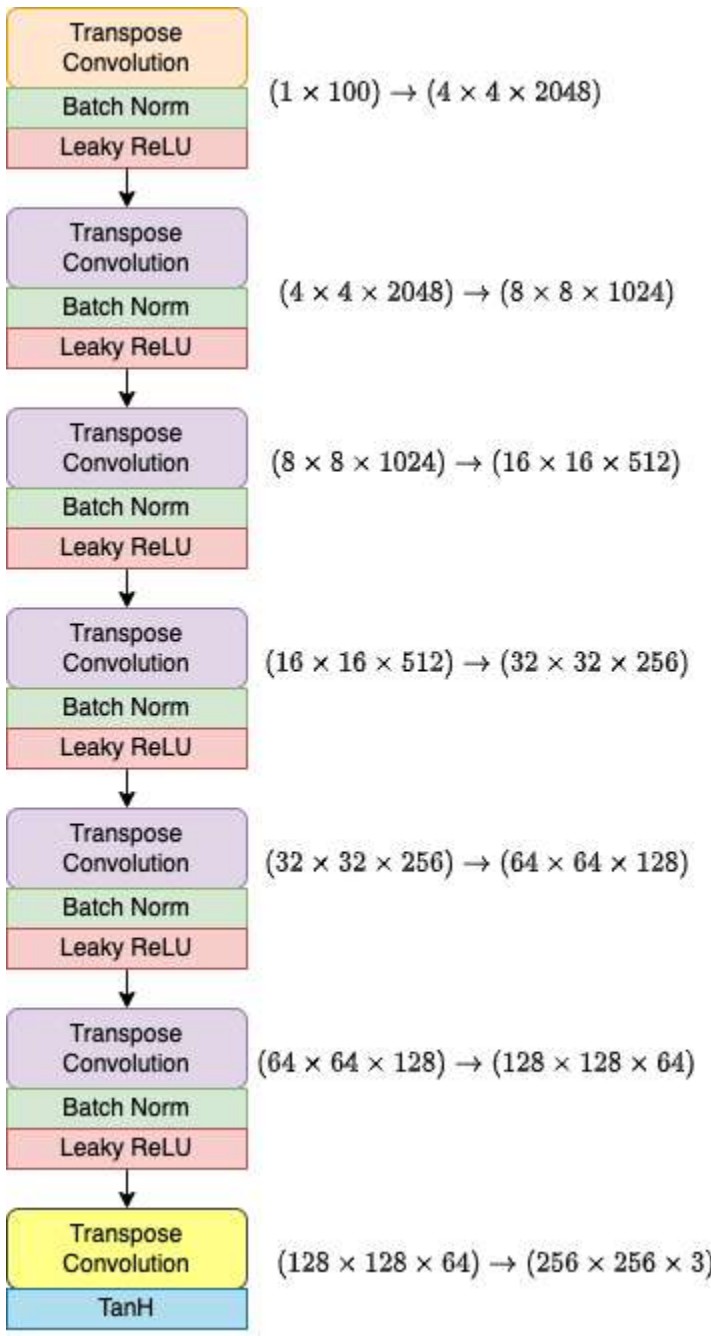

Figure 2: Generator Architecture (Image Dim 256)

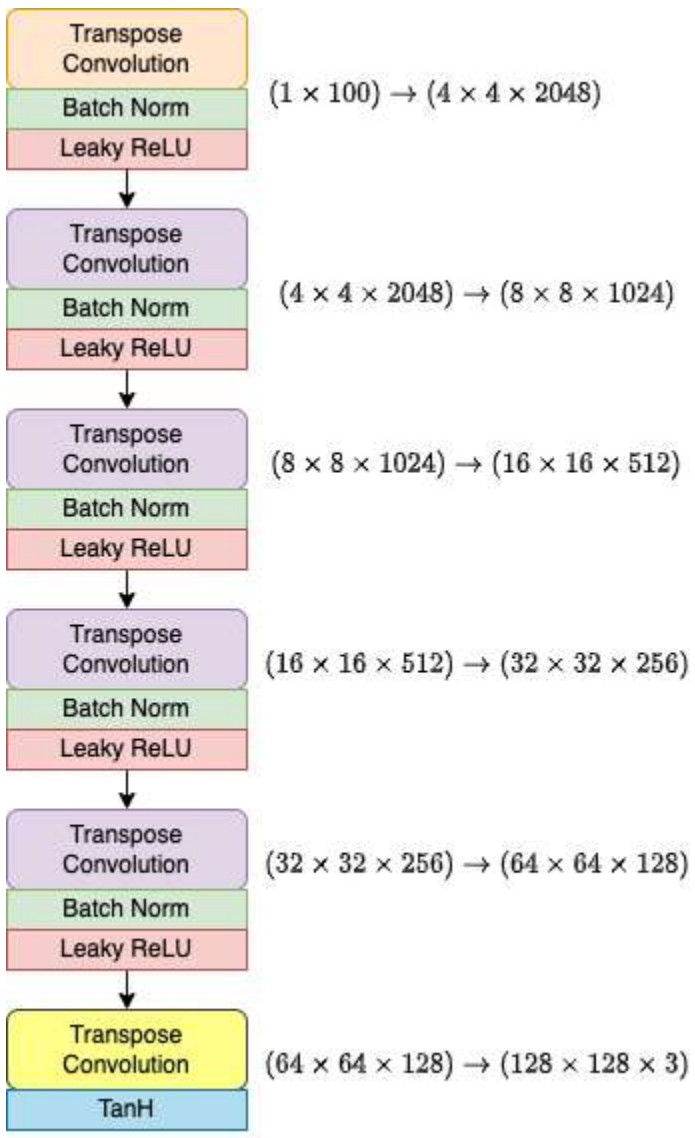

Figure 3: Generator Architecture (Image Dim 128)

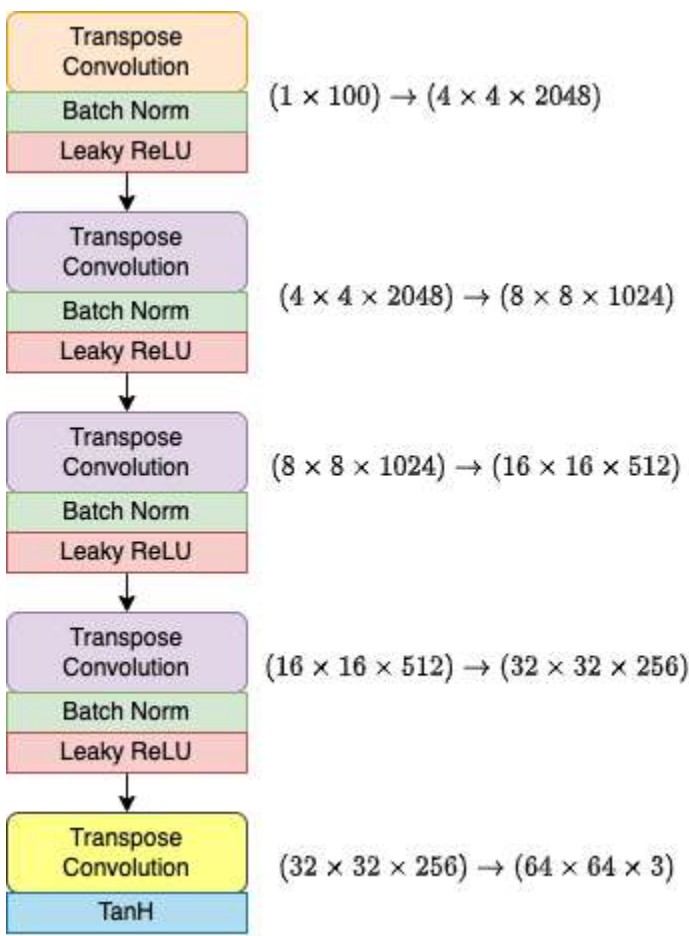

Figure 4: Generator Architecture (Image Dim 64)

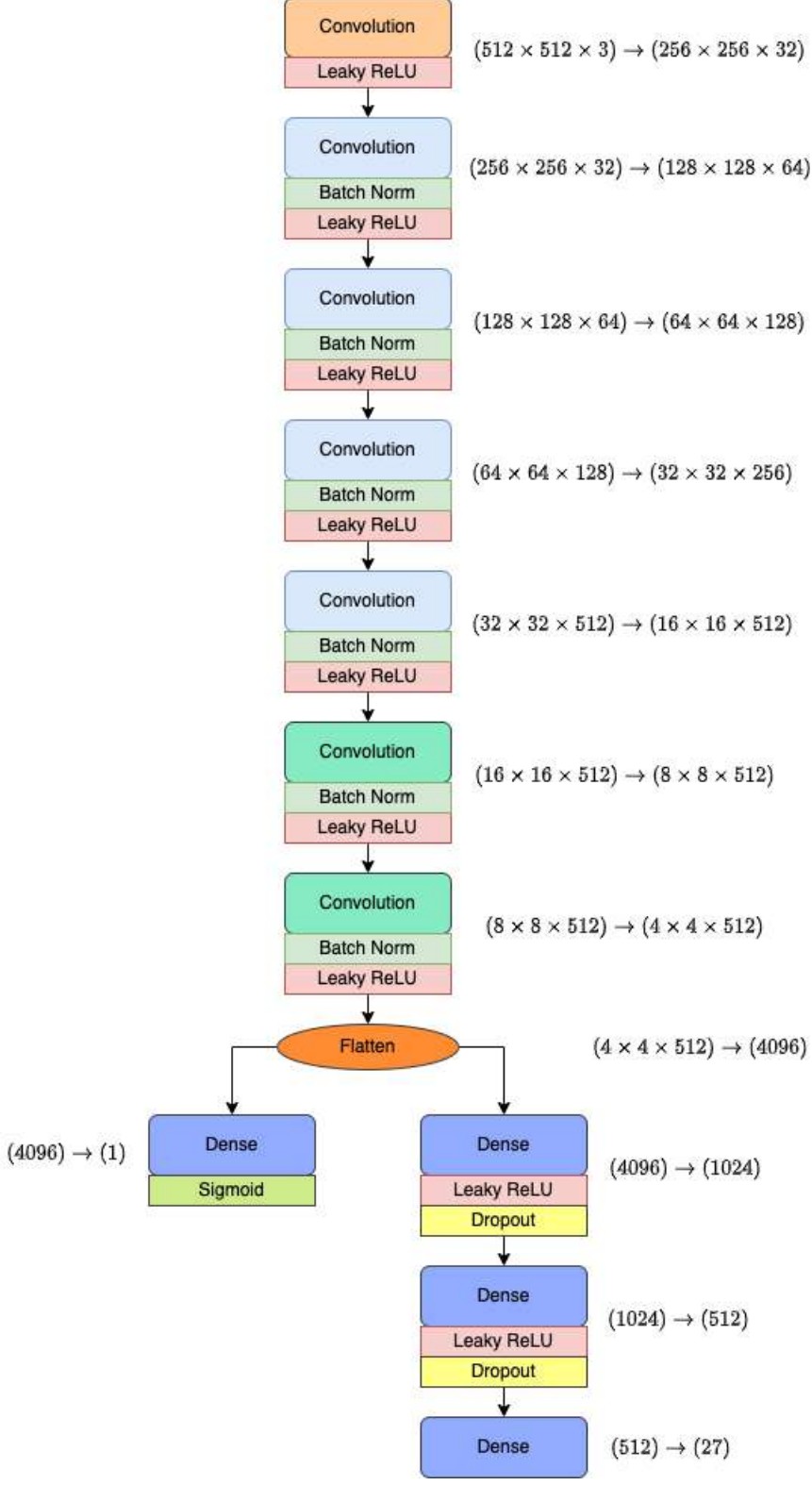

Figure 5: Discriminator Architecture (Image Dim 512)

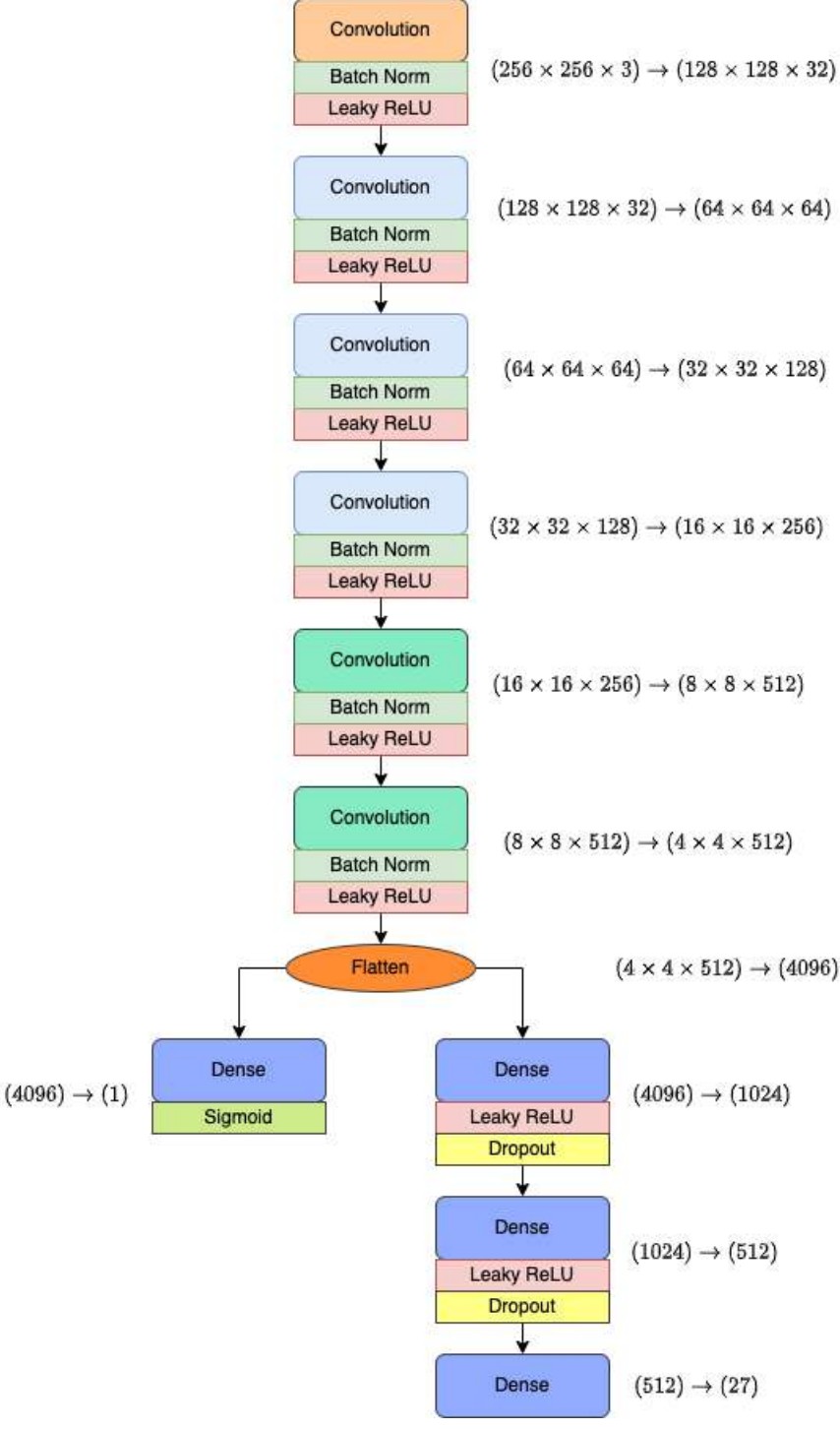

Figure 6: Discriminator Architecture (Image Dim 256)

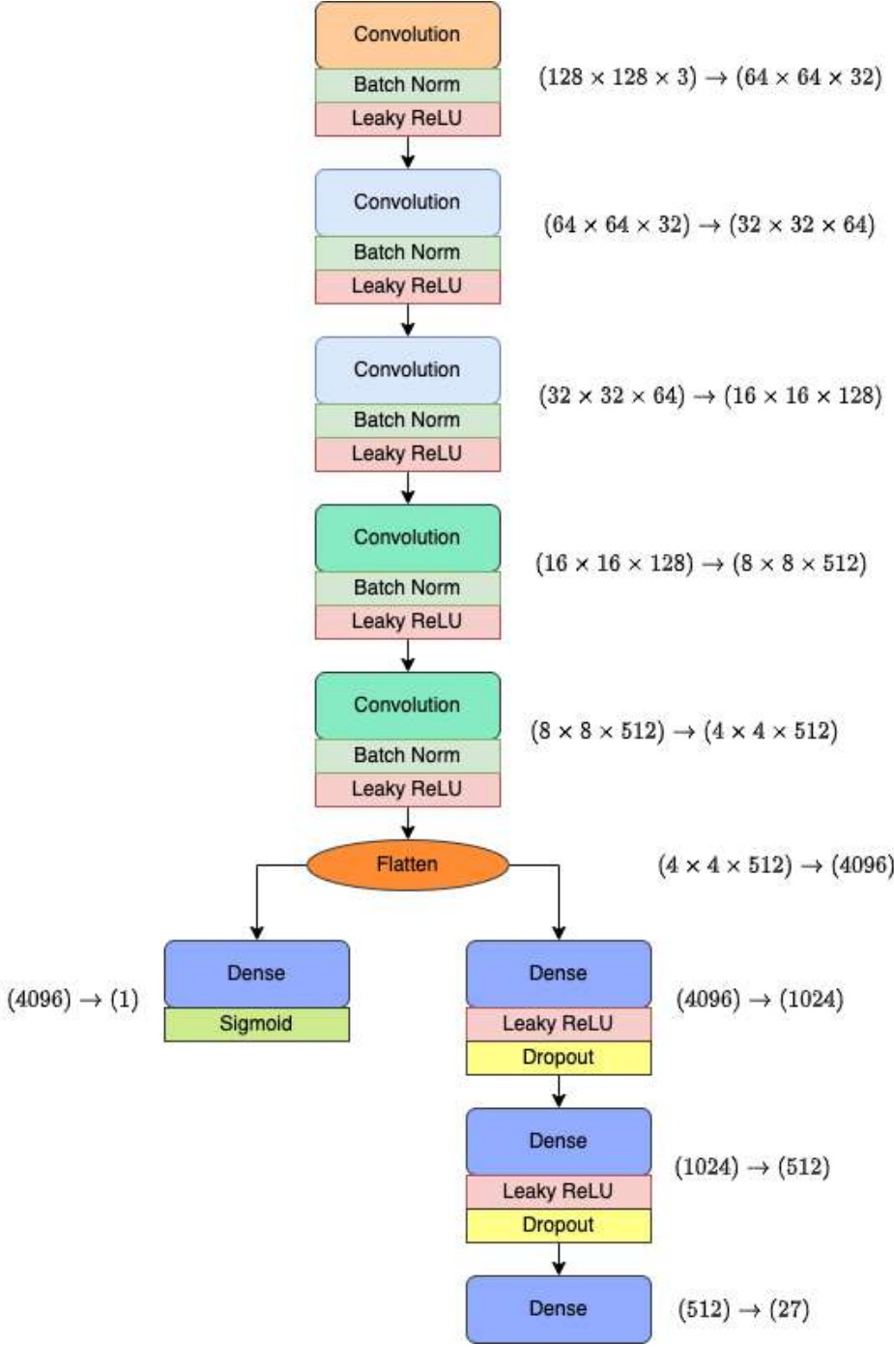

Figure 7: Discriminator Architecture (Image Dim 128)

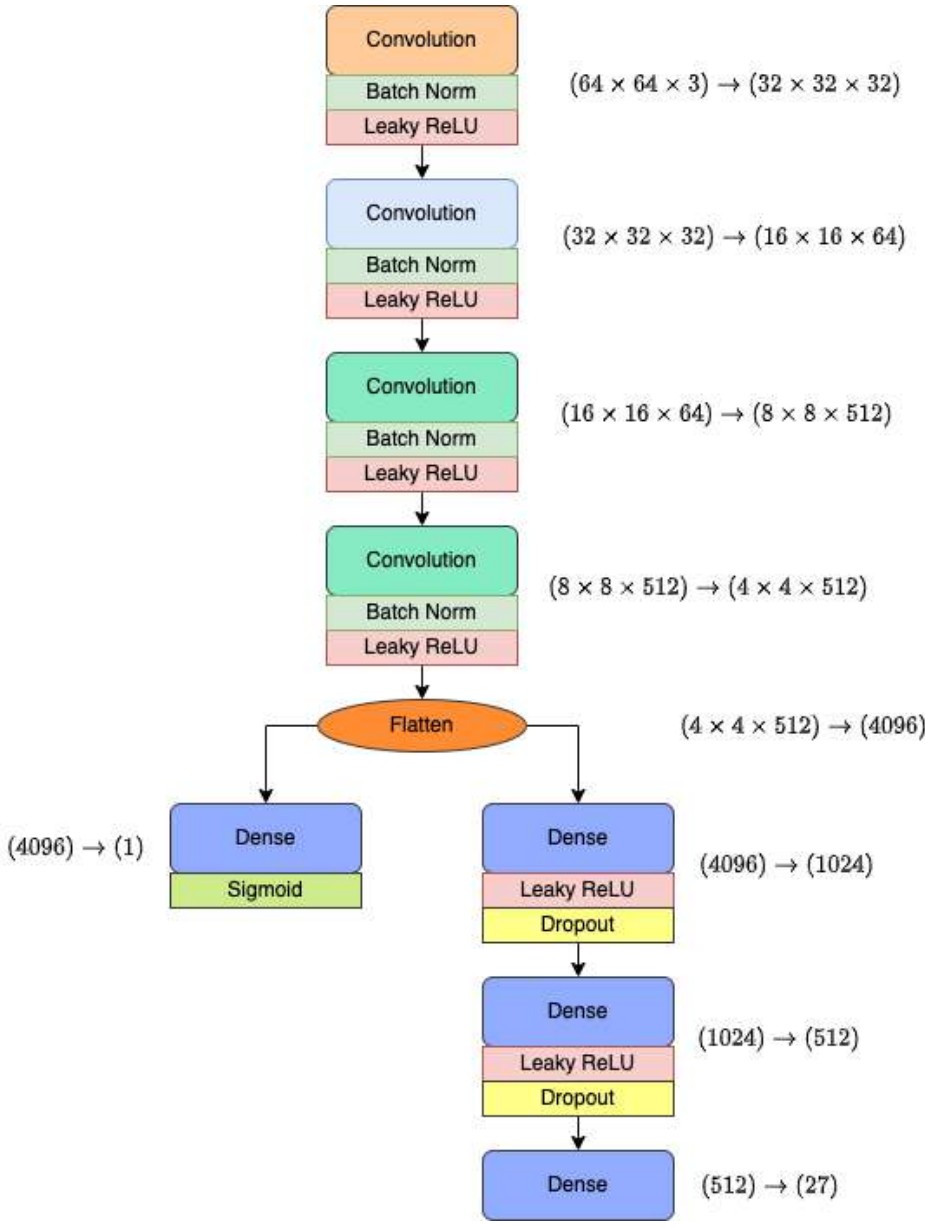

Figure 8: Discriminator Architecture (Image Dim 64)

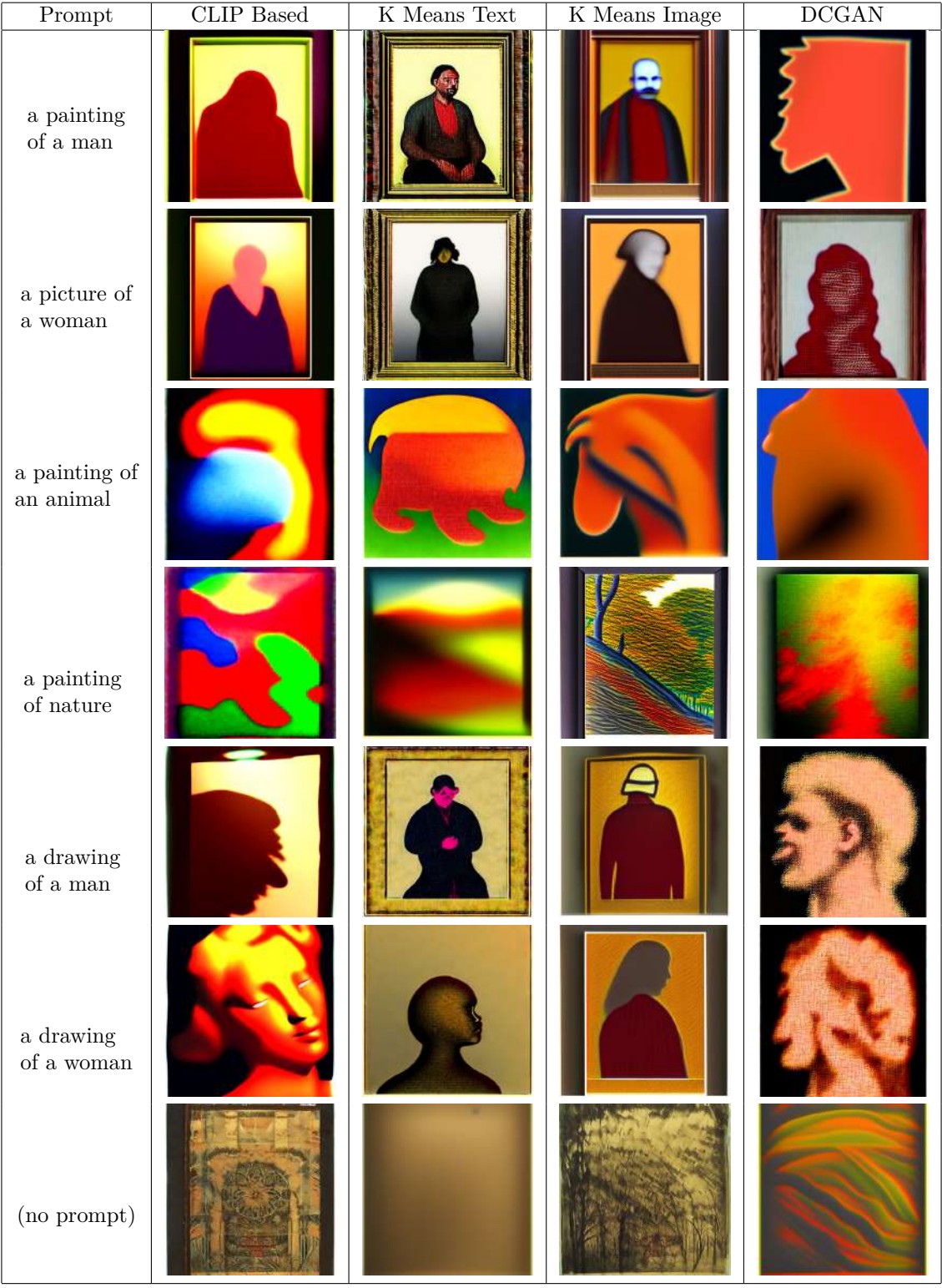

Table 5: Example Images (512)

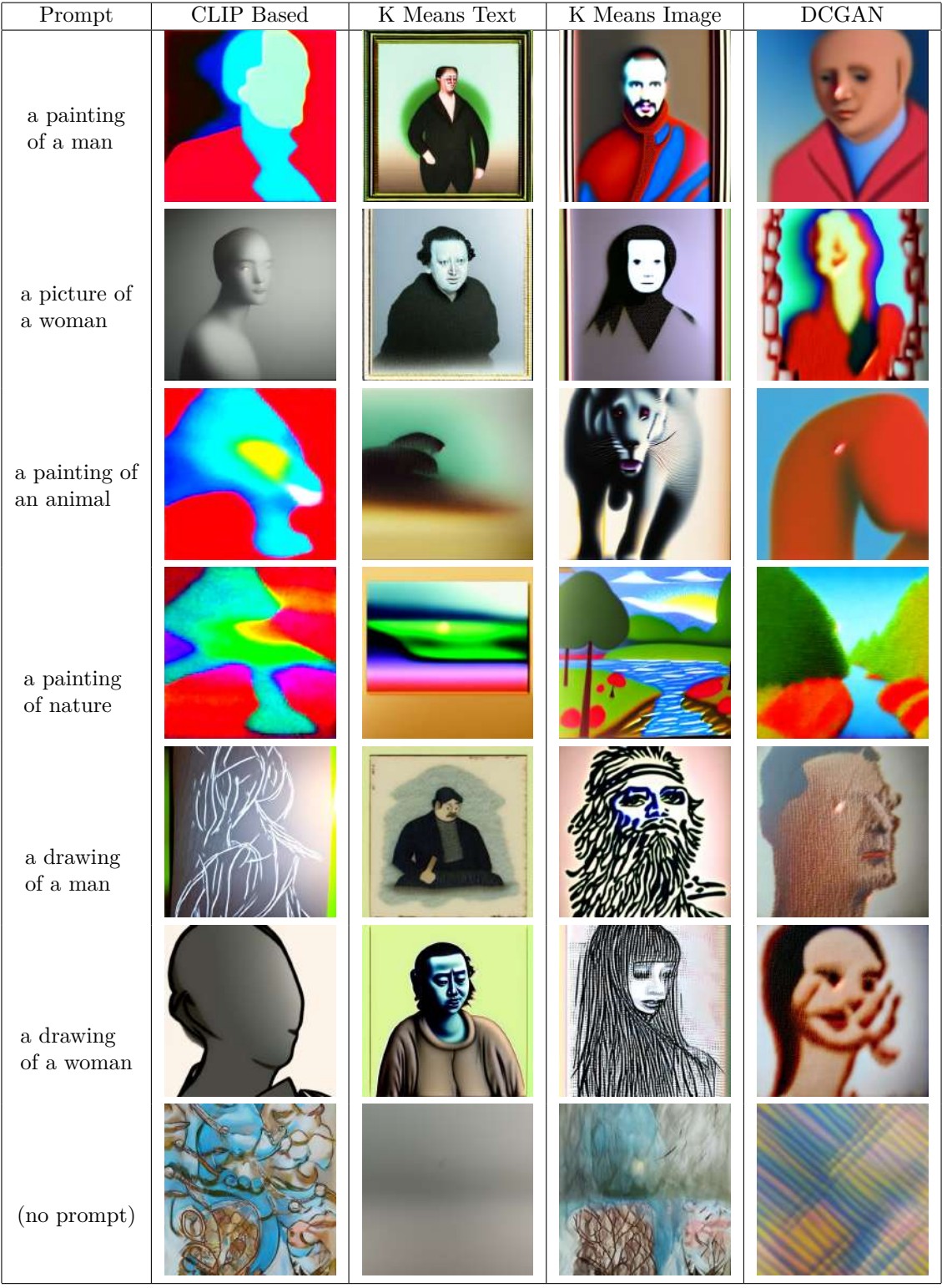

Table 6: Example Images (512)

| contemporary-realism | art-nouveau-modern | abstract-expressionism |
|---|---|---|
| northern-renaissance | mannerism-late-renaissance | early-renaissance |
| realism | action-painting | color-field-painting |
| pop-art | new-realism | pointillism |
| expressionism | analytical-cubism | symbolism |
| fauvism | minimalism | cubism |
| romanticism | ukiyo-e | high-renaissance |
| synthetic-cubism | baroque | post-impressionism |
| impressionism | rococo | na-ve-art-primitivism |

Table 7: Styles

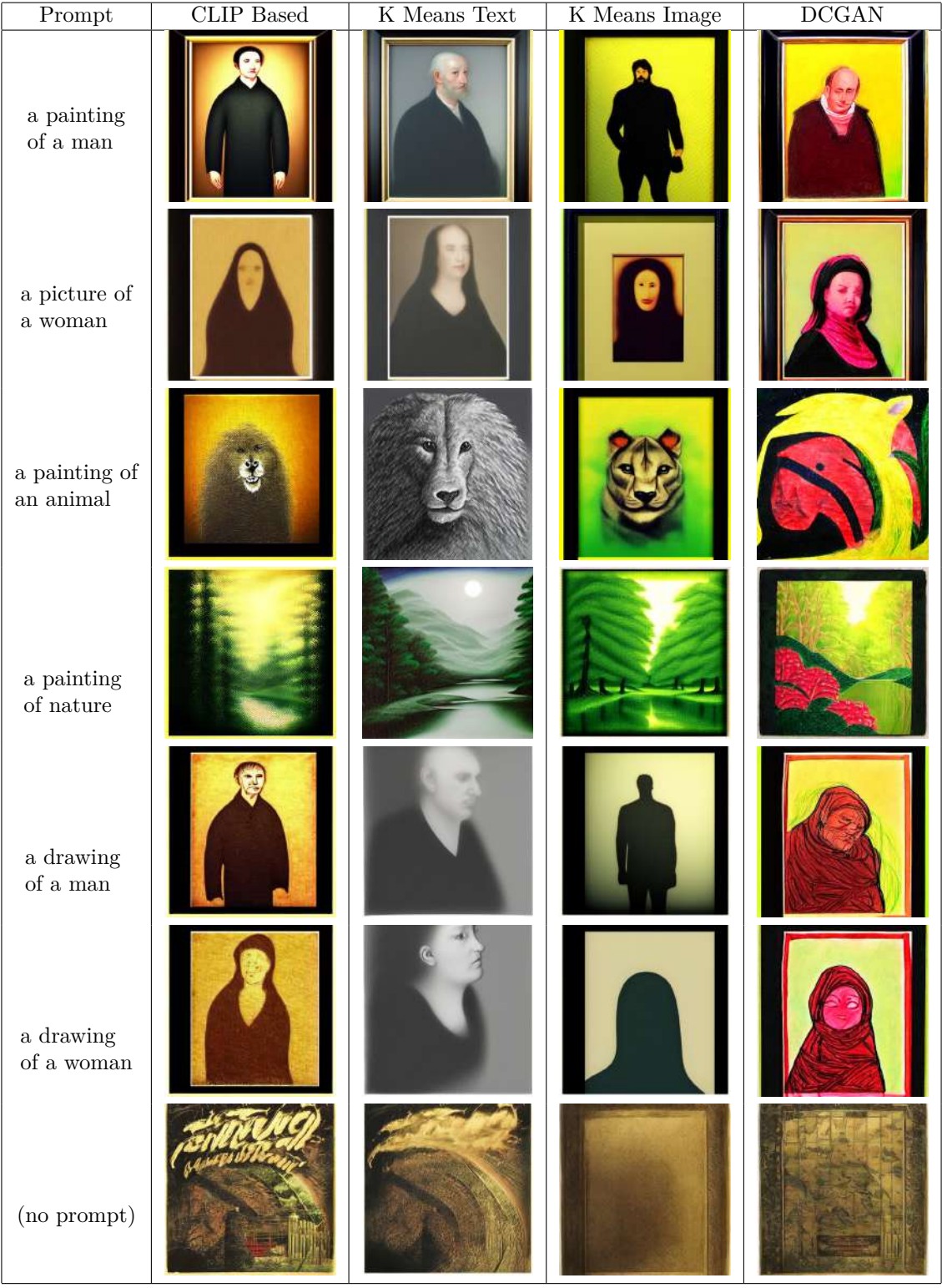

Table 8: Example Images (256)

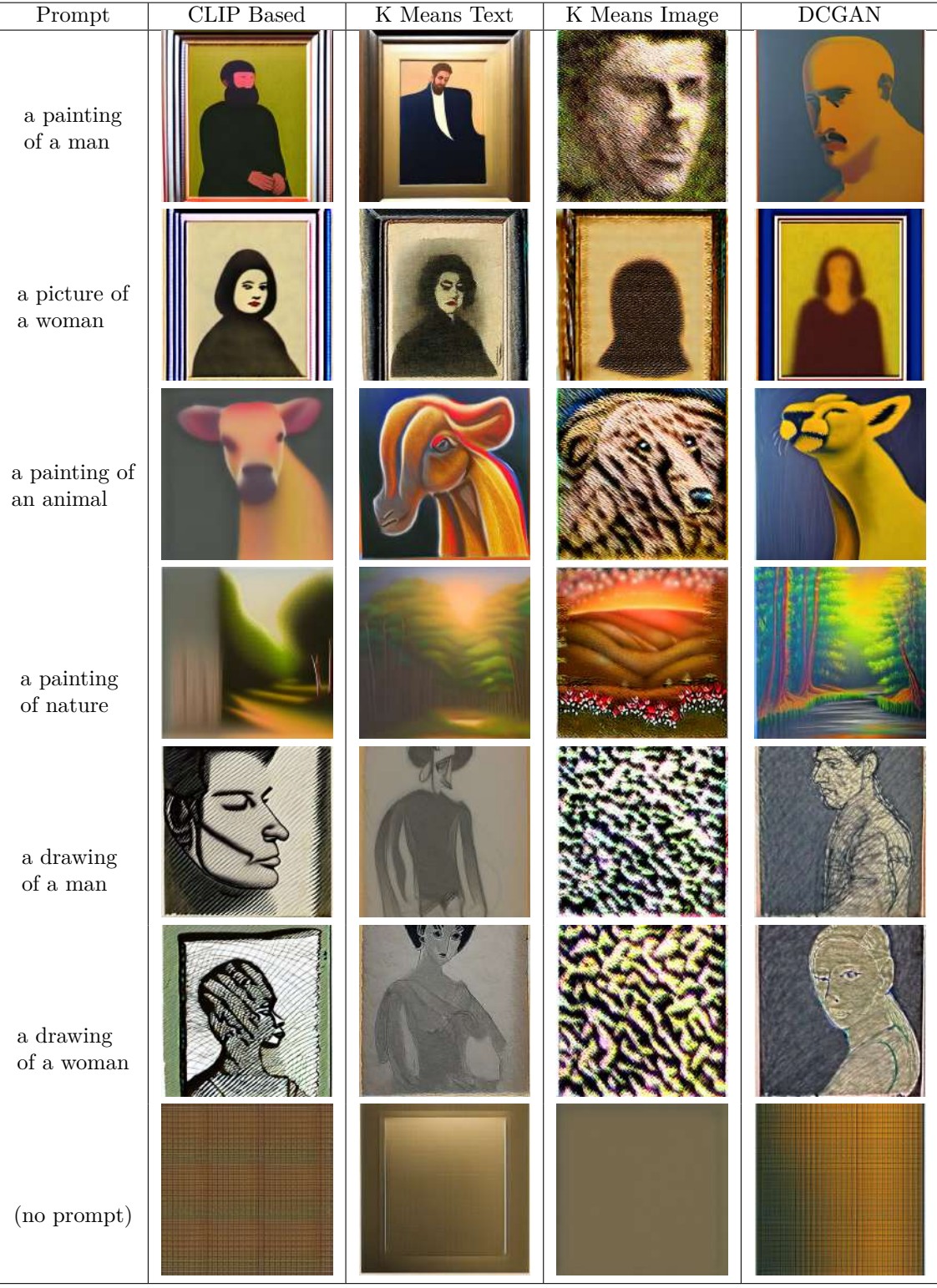

Table 9: Example Images (128)

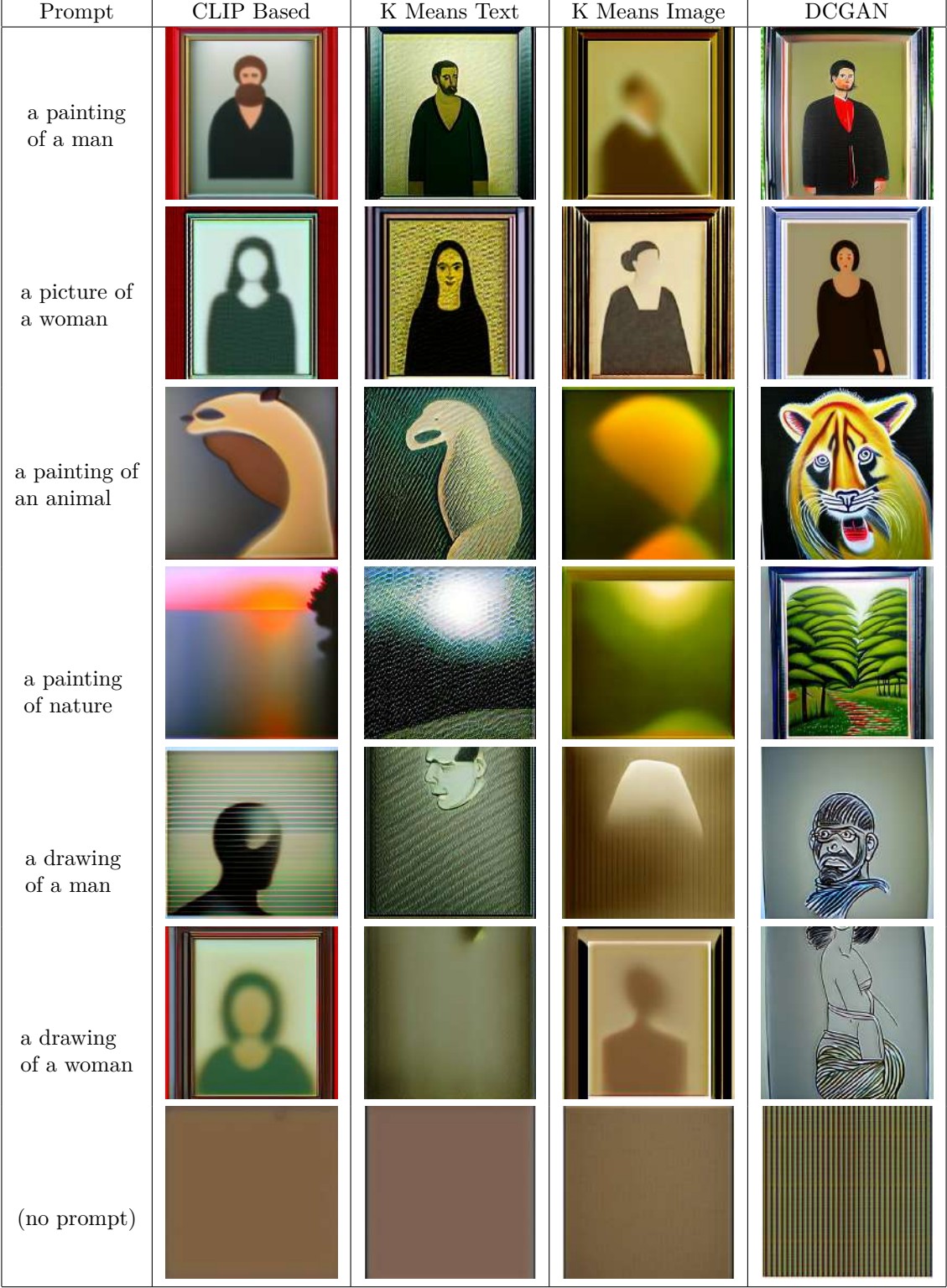

Table 10: Example Images (64)

| Model | AVA Score | Image Reward | Prompt Alignment |
|---|---|---|---|
| Diffusion- CLIP Based | 4.83 | -0.42 | 0.27 |
| Diffusion- K-Means Text Based | 4.91 | -0.64 | 0.27 |
| Diffusion- K-Means Image Based | 4.91 | -0.44 | 0.28 |
| Diffusion- DCGAN Based | 4.95 | -0.33 | 0.27 |

Table 11: Scores (256)

| Model | AVA Score | Image Reward | Prompt Alignment |
|---|---|---|---|
| Diffusion- CLIP Based | 4.47 | -0.74 | 0.27 |
| Diffusion- K-Means Text Based | 4.77 | -0.37 | 0.28 |
| Diffusion- K-Means Image Based | 3.78 | -1.36 | 0.25 |
| Diffusion- DCGAN Based | 4.66 | -0.14 | 0.29 |

Table 12: Scores (128)

| Model | AVA Score | Image Reward | Prompt Alignment |
|---|---|---|---|
| Diffusion- CLIP Based | 4.21 | -1.44 | 0.26 |
| Diffusion- K-Means Text Based | 4.36 | -1.00 | 0.27 |
| Diffusion- K-Means Image Based | 4.38 | -1.55 | 0.26 |
| Diffusion- DCGAN Based | 4.62 | -0.22 | 0.28 |

Table 13: Scores (64)

| Hyperparameter | Value |
|---|---|
| Epochs | 50 |
| Effective Batch Size | 8 |
| Batches per Epoch | 32 |
| Inference Steps per Image | 30 |
| LORA Matrix Rank | 4 |
| LORA $\alpha$ | 4 |
| Optimizer | AdamW |
| Learning Rate | 3e-4 |
| AdamW $\beta_1$ | 0.9 |
| AdamW $\beta_2$ | 0.99 |
| AdamW Weight decay | 1e-4 |
| AdamW $\epsilon$ | 1e-8 |

Table 14: DDPO Hyperparameters

| Hyperparameter | Value |
|---|---|
| Epochs | 100 |
| Batch Size | 32 |
| Optimizer | Adam |
| Learning Rate | 0.001 |
| Adam $\beta_1$ | 0.9 |
| Adam $\beta_2$ | 0.99 |
| Adam Weight decay | 0.0 |
| Adam $\epsilon$ | 1e-8 |
| Noise Dim | 100 |
| Wasserstein $\lambda$ | 10 |
| Leaky ReLU negative slope | 0.2 |
| Convolutional Kernel | 4 |
| Convolutional Stride | 2 |
| Transpose Convolutional Kernel | 4 |
| Transpose Convolutional Stride | 2 |

Table 15: CAN Hyperparameters

| Model | Hours | kgCO$_2$ |
|---|---|---|
| CAN (Image Dim 512) | 121.39 | 13.11 |
| Diffusion- CLIP Based (Image Dim 512) | 22.75 | 2.46 |
| Diffusion- K-Means Text Based (Image Dim 512) | 21.50 | 2.32 |
| Diffusion- K-Means Image Based (Image Dim 512) | 21.50 | 2.32 |
| Diffusion- DCGAN Based (Image Dim 512) | 21.44 | 2.32 |
| CAN (Image Dim 256) | 72.94 | 7.88 |
| Diffusion- CLIP Based (Image Dim 256) | 11.2 | 1.21 |
| Diffusion- K-Means Text Based (Image Dim 256) | 17.25 | 1.86 |
| Diffusion- K-Means Image Based (Image Dim 256) | 12.67 | 1.37 |
| Diffusion- DCGAN Based (Image Dim 256) | 8.53 | 0.92 |
| CAN (Image Dim 128) | 66.14 | 7.15 |
| Diffusion- CLIP Based (Image Dim 128) | 7.05 | 0.76 |
| Diffusion- K-Means Text Based (Image Dim 128) | 6.40 | 0.69 |
| Diffusion- K-Means Image Based (Image Dim 128) | 6.65 | 0.72 |
| Diffusion- DCGAN Based (Image Dim 128) | 6.24 | 0.67 |
| CAN (Image Dim 64) | 83.33 | 9 |
| Diffusion- CLIP Based (Image Dim 64) | 7.30 | 0.79 |
| Diffusion- K-Means Text Based (Image Dim 64) | 6.85 | 0.74 |
| Diffusion- K-Means Image Based (Image Dim 64) | 6.55 | 0.71 |
| Diffusion- DCGAN Based (Image Dim 64) | 6.14 | 0.66 |

Table 16: Training

| Model Component | Total Parameters | Trainable Parameters | Percent Trainable |
|---|---|---|---|
| Text Encoder | 34,0387,840 | 0 | 0% |
| Autoencoder | 83,653,863 | 0 | 0% |
| UNet | 866,740,676 | 829,952 | 0.1% |
| Generator (Image Dim 64) | 47,336,960 | 47,336,960 | 100% |
| Discriminator (Image Dim 64) | 13,691,612 | 13,691,612 | 100% |
| Generator (Image Dim 128) | 47,855,360 | 47,855,360 | 100% |
| Discriminator (Image Dim 128) | 14,347,228 | 14,347,228 | 100% |
| Generator (Image Dim 256) | 47,983,488 | 47,983,488 | 100% |
| Discriminator (Image Dim 256) | 15,920,604 | 15,920,604 | 100% |
| Generator (Image Dim 512) | 48,014,784 | 48,014,784 | 100% |
| Discriminator (Image Dim 512) | 20,115,932 | 20,115,932 | 100% |

Table 17: Parameter Counts

