# OpenReview forum: "Using Multimodal Foundation Models and Clustering for Improved Style Ambiguity Loss"
_TMLR — Rejected by TMLR_

### Review · Reviewer_t1wc · 2024-07-08

**Summary Of Contributions:**

The paper proposes to optimize/steer a text-to-image diffusion model at sampling time towards style ambiguity reward function. The style ambiguity loss function is the cross-entropy between a “ground-truth” uniform distribution and the “predicted” style distribution. It is inspired by similar loss function used in GAN-based models, Creative Adversarial Networks (CAN), Elgemmal et al. The paper makes two major contributions:

* Instead of training a GAN from scratch on a small dataset (WikiArt) to optimize style ambiguity, they rely on pretrained text-to-image model StableDiffusion. They finetune StableDiffusion during sampling, to steer the model to generate “style-ambiguous” outputs.

* Instead of training a classifier on art-specific inputs, they rely on pretrained models to obtain a distribution over art-styles.
They experiment with a) pretrained.CAN discriminator b) CLIP based classifier c) CLIP based K-Means, where clip image embeddings of the art images are clustered to obtain cluster centers. The predicted distribution of an image is just the softmax of the reciprocal of the distance from the clusters.

The paper generate 100 images and report AVA score, Image Reward and prompt similarity. They outperform the baseline (DCGAN based classifier) on AVA score and image reward.

**Audience:**

No

**Broader Impact Concerns:**

No such concerns

**Claims And Evidence:**

No

**Requested Changes:**

## Critical Experiments

* How much does the stable diffusion baseline (no finetuning) obtain in Table 2? These numbers are required to place the Table 2 metrics in context. This will help to understand the effect of optimizing style ambiguity on the other metrics
* What are the finetuning hyperparameters: (number of finetune steps, guidance parameters, etc)? All of these should be in the main section.
* There should be error-bars on Table 2. Please sample (100 samples) multiple times and report the standard errors.
* Is stable diffusion inherently biased towards certain styles as given by the CLIP classifer? It will be interesting to see the type of styles, stable diffusion is biased towards.
* How quickly do the “style ambiguity” reward converge? Please plot the reward as a function of finetuning steps.
* Just optimizing for style ambiguity may lead to a collapse. For example, the model can just generate an image, whose embedding is equidistant from all clusters but may not lie in the image manifold (For example: In Table 3, a painting of a man + clip-based model). Some sort of regularization may still be necessary to output a “natural-looking” image. For context, in the CAN paper, the generator is optimized with the “style ambiguity” loss plus the standard GAN loss $- \log D(G(z))$ which ensures that the images are still natural looking.
* The paper should have some sort of user studies comparing at least "surprising" or “ambiguity” metrics since this is what they optimize for as done in the CAN paper. These comparisons have to be done with a) Stable diffusion model b) DCGAN/ CAN model.
* What is the rationale for using a very small number of sampling steps in Table 4? Diffusion models need at least 256 steps to generate good samples.

## Critical Writing:
* “That does not require training a classifier” in the abstract seems a strong claim.The paper can be rephrased to “that uses pretrained foundational models”.
* Please expand the captions throughout

## Other suggestions (not critical):
* Section 3 can be divided into a background section and a method section for easier reading. The background section can be upto 3.1.4 and the remaining can be the method section.
* How are the number of clusters chosen for the KMeans algorithm?
* The paper describe the “Diffusion Simple Classifier Based” algorithm in Section 3.4.1 but does not use it anywhere.

**Strengths And Weaknesses:**

## Strengths

* The overall idea to steer models towards unique styles is quite interesting. The paper re-explores this idea in the context of foundational generative models which is quite useful.

* The paper is also quite easy to read. Most of the related background is provided, so the paper is self contained.


## Weaknesses

The experiments sections of the paper is quite weak. Based on the provided experiments, I find it difficult to conclude that the proposed algorithm or recipe is indeed capable of generating “style-ambiguous” outputs. See below for a list of detailed changes. With an improved experimental section, I'll be happy to reconsider my rating.

---

### Review · Reviewer_AnXb · 2024-07-14

**Summary Of Contributions:**

The authors contribute new formulations of creative style ambiguity loss, by using pre-trained models, and apply this loss to finetune a Stable Diffusion 2 unet. The authors demonstrate that out of the formulations, ones based on running K Means on CLIP's latent space are the strongest according to certain metrics and their own judgments. They then demonstrate that their finetuning approach does lead to very different outputs compared to the base model.

**Audience:**

Yes

**Claims And Evidence:**

No

**Requested Changes:**

Critical for scoring my recommendation:
1. A new evaluation that supports the authors' claims or a reworking of those claims.
2. Clarity on all missing or ambiguous detail in the paper.

I am happy to answer questions on either of the two of those. I do also feel specifically that a human subject study would greatly strengthen the paper, but I do see a path to an acceptance recommendation from me without one.

**Strengths And Weaknesses:**

# Strengths

The authors present a clear advantage to the existing creative style ambiguity loss in terms of allowing for their new losses to be applied to diffusion models and to not require additional training. The results do suggest that according to some metrics the K means-based creative style ambiguity losses are superior, which has implications for where others could use this approach in the future. The paper is overall well-written with clear descriptions of the technical contributions, which should also for easy replication by other readers.

# Weaknesses

This current paper has a number of weaknesses:
1. The results do not fully support the authors' current claims
2. There are ambiguities in the text
3. There is a mismatch in the amount of detail given to background information compared to relevant information to the authors' work

## Results and Claims

The authors make a number of claims in the current paper that are not fully supported by the results. The authors say their generated images have "higher quality than the generated samples of a diffusion model trained with the pre-existing GAN-based style ambiguity loss". They go on to say that their method "improves upon the past work aesthetically". These claims rely on readers accepting the AVA score and image reward as infallible metrics for evaluating quality and aesthetics. But this is not the case, as they are models simply approximating human judgements. There is no guarantee that a group of humans would necessarily agree with these metrics. Further, the difference in the metrics value is fairly low, making it difficult to tell if these differences are meaningful. The authors may be better served with a human subject study to better support these claims.

The authors also claim "Training models with stylistic ambiguity loss teaches them to be creative." This is a much larger claim to make and is even less supported by the current paper draft. While the authors gesture at a specific creativity definition in terms of novelty and utility, they do not explicitly identify that this is the measure that they are using in this paper. Even if it is, it is difficult for me to identify that any of the results support a claim that the finetuned Stable Diffusion 2 is more novel and equally useful. It is certainly different, but novelty is reliant on more than just difference. As the authors acknowledge in their discussion of p-creativity vs h-creativity, there's a conception of novelty historically that is missing from any of these measures. For example, it's quite possible that rather than leading to entirely novel outputs, the authors' finetuning has instead biased Stable Diffusion towards other parts of its learned latent space. If so, this wouldn't be novelty, but perhaps surprise (in Boden's conception of creativity).

I'd strongly encourage the authors to formally define what they mean by creativity in the paper and consider measures that could support their definition of creativity. For example, an analysis of the space of possible outputs (often referred to as the possibility space of a generator in computational creativity literature) could be an effective approach.

## Text Ambiguities

There are a number of instances of ambiguous text in the paper. It is unclear for 3.5 pages of the paper why Reinforcement Learning is being mentioned. It's unclear still to me at this point whether the DCGAN approach is meant to be an implementation of CAN and a baseline or whether it is a novel variant of CAN that the authors are proposing. The approach is referred to in both ways throughout the paper. The authors also refer to Stable Diffusion 2 as a VAE, even though the original authors only refer to it as like a VAE, and it is more commonly (and later in the paper) called a unet. It's also unclear which of the approaches are trained with the dataset described in 3.3. Some text suggests it is only the DCGAN, other text suggests it is all the models. It is also unclear if the given output images are hand-picked or randomly selected.

## Mismatch of Detail

The authors spend a great deal of time and detail covering background work, which is not all that relevant to this paper. I do not believe the authors need to give all of the equations for CAN, Diffusion, or the definition of a Markov Decision Process. These are not relevant to a reader's understanding of the authors' technical work. As the authors do for Stable Diffusion 2, I would suggest they simply cite prior work that covers these details for the interested reader. In comparison, as I have indicated in the last section, there are ambiguous or missing details which could be further clarified and expanded upon.

---

> ### Author Response · Authors · 2024-07-20
> **Request for clarification**
>
> Thank you for your response! I am currently reworking the paper. However, when you said, "For example, an analysis of the space of possible outputs (often referred to as the possibility space of a generator in computational creativity literature) could be an effective approach," I'm unsure what that would look like. If I understand correctly, the possibility space of a diffusion model would be every possible image that could be generated with the subset of prompts we want to study. What would the analysis of this (very large) space entail?

---

> > ### Comment · Reviewer_AnXb · 2024-07-20
> > **Re: Request for clarification**
> >
> > I'm happy to clarify! You are entirely right that the possibility space of a generator is itself a theoretical concept and not one that is (typically) measured. Instead, the typical solution is an expressive range analysis [1]. In more ML-y parlance, by sampling an arbitrarily large number of times from the model and then mapping them to some smaller space (such as very a TSNE or DBSCAN clustering approach), it's possible to visualize the differences between generators. For example in [2]. There are many other ways to approximate the possibility space of a generator of course, I am not saying this is the only option but it is a common one in the computational creativity research field.
> >
> > 1. Smith, Gillian, and Jim Whitehead. "Analyzing the expressive range of a level generator." Proceedings of the 2010 workshop on procedural content generation in games. 2010.
> > 2. Summerville, Adam. "Expanding expressive range: Evaluation methodologies for procedural content generation." Proceedings of the AAAI Conference on Artificial Intelligence and Interactive Digital Entertainment. Vol. 14. No. 1. 2018.

---

### Review · Reviewer_L76m · 2024-07-16

**Summary Of Contributions:**

This work proposes a method for text-to-image focusing on ambiguity in generation styles by employing a pre-trained text-image model, i.e., CLIP, as a classifier in the GAN style generation. The motivation is to generate creative images, but its criteria is not clear. This work introduces two classifiers based on CLIP, one is to cluster the styles by the CLIP representation and the other using k-means.

**Audience:**

No

**Broader Impact Concerns:**

None.

**Claims And Evidence:**

No

**Requested Changes:**

- Basically, this manuscript sounds like a report on applying various classifiers to the GAN-like image generation focusing on styles, but we would expect what is the motivation of working on art image generation, and what are challenges under the task. Then, this manuscript should explain how and why the proposed approach resolve the problems.

**Strengths And Weaknesses:**

Strengths:
- The use of CLIP as a classifier under GAN style image generation seems to be a contribution and this work has investigated two approaches, one for explicitly classifying image according to the similarity of the pre-defined classes in WikiArt, and the other leveraging k-means to search for similar representations.
- Experiments show variances in the results, but in general, the proposed approach with k-means seem to be better than ohters.

Weaknesses:
- The motivation of this work is not clear. This work needs to describe the challenges in the styled image generation focusing on artistic images when compared with image generation in general or other domains.
- The motivation of the proposed methods are not clear. Since the motivation on working on art domain is not clear, it is unclear what challenges the proposed methods are trying to resolve. I'd expect any hypotheses which might indicate the success or failure of the proposed approach.
- Similarly, the analysis is very weak and it is not clear what issues this work is trying to address.

---

### Decision · Action_Editor_gzsv · 2024-08-21

**Recommendation:** Reject

**Comment:**

The reviewers could not provide any further feedback as there has been little response from the authors. Given the unanimous response from the reviewers, I propose to reject the paper. While the idea looks promising, addressing the comments could strenghen the paper.

**Audience:**

As Reviewer t1wc notes the overall idea to steer models towards unique styles is quite interesting. However, it would be interesting if there is better backup of claims in terms of both the experiments and metrics, as detailed by  t1wc, AnXb. the authors need to also address the concerns about the motivation as outlined by  L76m.

**Claims And Evidence:**

The reviewers have highlighted a number of concerns related to the claims and evidence.

1)  t1wc suggests that the stable diffusion baseline without finetuning should be present in Table 2 to provide the baseline

2) a number of details is unclear or ambiguous as highlighted by all of the reviewers